# The Effect of Additional Treatment with Empagliflozin or Semaglutide on Endothelial Function and Arterial Stiffness in Subjects with Type 1 Diabetes Mellitus—ENDIS Study

**DOI:** 10.3390/pharmaceutics15071945

**Published:** 2023-07-14

**Authors:** Maja Preložnik Navodnik, Andrej Janež, Ivan Žuran

**Affiliations:** 1Department of Angiology, Endocrinology and Rheumatology, General Hospital Celje, Oblakova ul. 5, 3000 Celje, Slovenia; 2Department of Endocrinology, Diabetes and Metabolic Diseases, University Medical Centre Ljubljana, Zaloška Cesta 7, 1000 Ljubljana, Slovenia; andrej.janez@kclj.si

**Keywords:** endothelial function, empagliflozin, semaglutide, type 1 diabetes mellitus, flow-mediated dilation, forearm blood flow, arterial stiffness

## Abstract

We investigated the effect of additional treatment with newer antidiabetic drugs on endothelium function and arterial stiffness in subjects with type 1 diabetes mellitus (T1DM) without cardiovascular diseases. A total of 89 participants, all users of CGMS (continuous monitoring glucose system), were randomized into three comparable groups, receiving empagliflozin (E; *n* = 30), receiving semaglutide (S; *n* = 30), and a control group (C; *n* = 29). At baseline and 12 weeks post treatment, we measured FMD (brachial artery flow-mediated dilation) and FBF (forearm blood flow as reactive hyperemia assessed with strain gauge plethysmography) as parameters of endothelial function, as well as pulse wave velocity (PWV) and peripheral resistance as parameters of arterial stiffness. Improvement in FMD was significant in both intervention groups compared to controls (E group 2.0-fold, *p* = 0.000 and S group 1.9-fold, *p* = 0.000), with no changes between those two groups (*p* = 0.745). During the evaluation of FBF, there were statistically insignificant improvements in both therapeutic groups compared to controls (E group 1.39-fold, *p* = 0.074 and S group 1.22-fold, *p* = 0.701). In arterial stiffness parameters, improvements were seen only in the semaglutide group, with a decline in peripheral resistance by 5.1% (*p* = 0.046). We can conclude that, for arterial stiffness, semaglutide seems better, but both drugs positively impact endothelial function and, thus, could also have a protective role in T1DM.

## 1. Introduction

Diabetes is not only a metabolic disease; it is also considered a vascular disease because of its effect on macro- and microcirculation. The evidence from cardiovascular outcome trials on the benefits of GLP-1 (glucagon-like peptide 1) agonists and SGLT 2 (sodium–glucose cotransporter 2) inhibitors has significantly modified treatment strategies for individuals with type 2 diabetes (T2DM) [1,2,3,4].

A multifactorial approach beyond the glucose-lowering effect of these drugs is crucial in routine clinical practice to reduce cardiovascular risk in individuals with T2DM. In addition to T2DM, T1DM is associated with higher mortality and cardiovascular disease (CVD) risk than the general population [5,6]. Hyperglycemia appears to have a more profound effect on cardiovascular risk in T1DM, while, in T2DM, achievement of treatment targets for all recognized risk factors is crucial [7,8]. A pivotal role in cardiovascular health and disease is held by the vascular endothelium, which is a multifunctional organ responsible for regulation of vascular tone and integrity [9]. The associated endothelial dysfunction is now accepted as a reliable predictor of cardiovascular disease [10]. The assessment of endothelium function is based on endothelium-dependent vasodilation as a response to stimuli, which increases the production of endothelium-derived nitric oxide (NO). Such stimuli include increased shear stress from increased blood flow, (postischemic or driven by receptor-dependent agonists, such as acetylcholine). NO relaxes vascular smooth muscle and is also a potent antioxidant and a regulator of local and systemic redox status [11]. The diagnostic modalities include invasive and more frequently used noninvasive techniques such as flow-mediated dilatation of the conduit brachial artery using vascular ultrasound [12,13], local vasodilation by venous occlusion plethysmography [14], and microvascular blood flow by laser Doppler flowmetry [15], while arterial pulse wave analysis or pulse amplitude tonometry evaluate arterial stiffness, which is also highly dependent on fixed structural features of the vascular wall, including the degree of fibrosis and calcification [16]. Protocols have been largely standardized, and this has resulted in reproducible measurements.

Endothelial dysfunction in T1DM is an early phenomenon that is relatively common even in adolescents with recent onset of diabetes, independent of age, smoking, hypertension, or hyperlipidemia [17]. Higher HbA1c levels in patients with T1DM were associated with more pronounced endothelial dysfunction in the whole population (β = −0.20; *p* < 0.05) [18]. A moderation of the impact of BMI on endothelial dysfunction was also found in all individuals with T1DM; the difference in mean BMI between children/adolescents with type 1 diabetes and healthy controls was positively associated with endothelial function but only when analyzing the macrocirculation [18]. Recently, it has been postulated that the main cardiovascular risk factor is not only chronic hyperglycemia or other traditional risk factors, but also frequent hypo- and hyperglycemia episodes that accompany the disease daily course, i.e., excessive glycemic variability [19,20]. FMD improvement was found (10.9% to 16.6%, *p* < 0.005) after switching individuals with T1DM to real-time continuous glucose monitoring (RT-CGM) [21]. The literature has shown that statins and ACE inhibitors have beneficial effects on FMD [22,23].

To date, there are few published studies that have looked at the effect of newer antidiabetic agents on endothelial function, most of which relate to SGLT 2 inhibitors and only a few of which relate to GLP 1 agonists; even among these, semaglutide was not included as an interventional drug. Most studies have been performed in T2DM [24,25,26], with only two being performed in T1DM [27,28]. A meta-analysis of eight studies on the effect of SGLT2 inhibitors on FMD showed that the ability of SGLT-2i to improve FMD was significant compared to the control group [29]. A systematic review and meta-analysis, conducted by Batzias et al., included a total of 26 eligible studies in which SGLT-2 inhibitors significantly improved FMD (pooled MD 1.14%, 95% CI: 0.18 to 1.73, *p* = 0.016), but not GLP-1 RA (pooled MD = 2.37%, 95% CI: −0.51 to 5.25, *p* = 0.107) [26]. GLP-1 RA (pooled MD = −1.97, 95% CI: −2.65 to −1.30, *p* < 0 001) also significantly decreased PWV [26]. Those studies suggested that improved vascular function was likely associated with empagliflozin-mediated glycemic and nonglycemic actions, such as weight loss and volume contraction. The aim of the present study was to observe and compare metabolic and endothelial function-related effects of newer therapies with good cardiovascular outcomes (empagliflozin and semaglutide) as adjuvant therapies to insulin in well-controlled individuals with T1DM. Most research takes place in T2DM, as the prescription of newer therapies is off-label with T1DM. On the basis of the results from the study, we aimed to determine which parameters of endothelial function can be influenced by the introduction of a single drug, while also evaluating their potential for use in T1DM.

## 2. Methods

### 2.1. Study Design and Subject Selection

The ENDIS study (endothelium dysfunction assessment study) was a 12 week intervention, prospective, randomized, single-center, controlled clinical study with 92 recruited individuals with T1DM on insulin treatment either with MDI (multiple daily injections) or CSII (continuous subcutaneous insulin delivery), all using CGMS (continuous glucose monitoring systems).

They were randomized according to their demographic characteristics and accompanying risk factors into three comparable groups: Group E with empagliflozin 10 mg as addon to insulin, Group S with semaglutide 1× weekly s.c (titration from 0.25 to 1 mg according to the recommendations from Smpc or to maximal tolerable dose), and control C group treated only with insulin. The lower dose of empagliflozin was chosen in the study design in view of the equal efficacy and benefit on cardiovascular outcomes of both 10 and 25 mg doses in large CVOT studies performed in T2DM [1], as well as to reduce the possibility of DKA for safety reasons. We also took into account the EMA’s decision in the case of dapagliflozin, which was the only SGLT 2 inhibitor with short-term approved indication for the treatment of T1DM; the approved dose was lower.

We did not enroll subjects who were on metformin to reduce potential confounding on endothelial function measurements [27,28], and they were naïve of study drugs. All were users of CGMS for at least 3 months (real-time Dexcom 6 or Guardian 3 or 4; a few of them were users of flash CGMS Libre 1—in this case, they were switched to RT (real-time) CGM 2 weeks before inclusion). Exclusion criteria were a history of any manifested atherosclerotic disease, renal insufficiency (creatinine clearance <60 mL/min), known active malignant disease, chronic systemic connective tissue disease or chronic wounds, poorly controlled diabetes—HbA1C > 9.0%, BMI < 22, and all contraindications according to Smpc of inclusion drugs. The study protocol was approved by the National Medical Ethics Committee of Slovenia and all subjects gave written informed consent prior to participating in any study procedures. The study was registered at Clinicaltrials.gov with No. NCT05857085.

### 2.2. Statistical Analysis

The GPower 3.1.9.2 software and the Wilcoxon signed-rank test for paired samples were used to calculate the sample size assuming error probability of 5% (α = 0.05) and desired statistical power >80%. The effect size (d) was determined on the basis of the average %FMD from the scientific literature [27,29,30]. Considering a correlation coefficient between groups of r = 0.5, we calculated an effect size of d = 0.54. The sample size calculation showed that we needed 31 individuals per group to provide at least 81.33% statistical power.

The obtained results of the study were analyzed using the IBM SPSS Statistics 25.0 software package (IBM, Armonk, NY, USA) and the R 4.0.3 programming environment (R Core Team 2020, Vienna, Austria). Continuous variables were tested using the Shapiro–Wilk normality test in order to assess the normality of continuous data distribution. Categorical data were analyzed using the chi-square test. Continuous variables between groups were analyzed using Mann–Whitney U-Test or Kruskal–Wallis H-test with all pairwise comparison. We also performed statistical analysis using generalized linear models (GLMs) in order to adjust statistical results for selected confounding covariates. Correlations were assessed using Spearman’s rank correlation. All statistical tests were two-tailed, and a *p*-value ≤ 0.05 was considered statistically significant.

### 2.3. Study Protocol

Prior to both study visits, participants were asked to avoid vitamin supplementation or extensive physical activity, while women were asked to come in the same period of menstrual cycle. All participants were fasting and without caffeine consumption, and they were also asked to avoid smoking and to come healthy without any cold symptoms to limit influences on vascular endothelial function. Visits were conducted early in the morning. Anthropometric measurements were taken (height, weight, and waist circumference in metric units). Blood was obtained for laboratory test measurements, and blood pressure was recorded using an automated sphygmomanometer (measured in triplicate and averaged after 10 min rest). Data from RT CGM were downloaded for a period of 2 weeks before study drug initiation, and 2 weeks before the control visit. Participants were educated to reduce the insulin dose as needed. In the case of randomization in group E, they also received a ketone measuring device (Wellion) and were educated about the possibility and prevention of euglycemic ketoacidosis occurrence.

### 2.4. Endothelial Function Assessment

All measurements were performed after at least 15 min of rest in a supine position in a quiet and temperature-controlled room. Each method was performed with the same investigator, to eliminate interobserver variability, and methods were well validated and used in the past with investigators in the outpatient clinic at General Hospital Celje.

### 2.5. Brachial Artery Fow-Mediated Dilation (FMD) Assessment

Ultrasound assessment of brachial artery reactivity was performed in accordance with the guidelines of the International Brachial Artery Reactivity Task using the General Electric Logiq S7 Expert/Pro ultrasonic device, with a 9L-D probe and a frequency band of 3.1–10 MHz in B mode, which was fixed on the probe holder to avoid movements [10,13]. After measurements of the baseline artery diameter, the sphygmomanometer cuff on the forearm was inflated to 50 mmHg above systolic blood pressure (which was measured on the contralateral arm) and kept for 4 min. Brachial artery diameter (BMD) was measured and recorded again during reactive postischemic hyperemia for a duration of 2 min after cuff deflation. Maximal values were used for calculation of flow-mediated dilation and defined as the percentage change in artery diameter relative to the baseline diameter (%FMD = [BMDish − BMDbaz]/BMDbaz). As our population did not have signs of macrovascular diseases, we omitted the control test for nonendothelial-dependent vasodilation measured as the vasodilator response to 0.4 mg sublingual nitroglycerin, as defined in the reference protocol [13], because it is often accompanied by a headache.

### 2.6. Forearm Blood Flow (FBF) Reactive Hyperemia Assessment

For measurements of forearm blood flow, we used a strain gauge plethysmograph (model EC5R, D.E. Hokanson, Inc., Bellevue, WA, USA) according to protocol, as described by Higashi [10,14]; again, the aim was to establish postischemic reactive hyperemia. A strain gauge was attached to the upper part of the forearm and connected to a plethysmography device, where the flow curve was recorded on thermolabile paper. A wrist cuff was inflated to a pressure of 50 mmHg above the systolic blood pressure to exclude hand circulation from the measurements 1 min before and throughout the measurements of FBF. The upper arm congesting cuff was inflated to 50 mmHg for 7 s in each 15 s cycle to occlude venous outflow from the arm using a rapid cuff inflator. Baseline measurements were performed for 1 min. The cuff was inflated to 200 mmHg on the upper arm for 4 min. Before releasing the flow in the upper arm, we inflated the wrist cuff for a time of FBF measurements. The reactive postischemic hyperemia was achieved after cuff release, and FBF was measured for 3 min (12 measurements) as described above. FBF was later calculated as the area under the FBF curve and expressed as mL per 100 mL of forearm tissue volume per minute. We calculated two values: forearm blood flow in the first minute where the highest FBF was detected, and total FBF in a 3 min interval, measured throughout the whole hyperemic response and recovery toward the baseline blood flow, and was expressed as the mean of all 12 measurements. Reactive hyperemia (RH) was calculated separately for 1 and 3 min intervals as the ratio of the postischemic increment and basal tissue flow before cuff inflation ([FBFish – FBFbaz]/FBFbaz).

### 2.7. Arterial Stiffness Assessment

For each individual, we performed four measurements taken at a time interval of 2 min using a Schiller BR 102 device, which contains an algorithm and mathematical model in the inbuilt software using estimations of the amplitude and time difference of the first and second waves for pulse wave velocity (PWV). Wave reflection occurs at sites of impedance mismatch, often branch points, and it is usually quantified by determining the augmentation index (AIx), which represents the difference between the first and the second systolic peaks [10,16]. The morphology of the reflected second wave is determined by arterial stiffness. PWV is calculated as the ratio of the distance (Dx) between two arterial sites, and the time delay (Dt) of the pulse between these sites is expressed in m/s. The augmentation index (Alx) is determined as the arterial wave reflection magnitude [(reflected/forward wave amplitude) × 100%], AIx75 was automatically calculated to adjust the AIx for a heart rate of 75 beats/min using the formula AIx75 = ([heart rate − 75] × 0.39) + AIx.

Device calculation was also used to determine peripheral vascular resistance in blood flow as PR = (change in pressure across the circulatory loop)/flow and expressed in mmHg·L^−1^·min^−1^ [31].

## 3. Results

### 3.1. Participant Disposition, Demographics, and Baseline Characteristics

A total of 91 participants were enrolled in the study and randomized; among these, two participants withdrew from the study—one because of the side-effects from a study drug (nausea, vomiting), and the other one from the control group due to a personal reason. Therefore, 89 participants completed the study. Participants in group E took empagliflozin (*n* = 30) while participants in group S took semaglutide (*n* = 30) as an addon to insulin, while the control group therapy was unchanged (*n* = 29). Demographic and baseline characteristics *(*Table 1*)* were similar between groups (age, sex disposition, diabetes duration, smoking status, body mass index (BMI), weight, waist, and eGFR) except for baseline HbA1C, which was higher in the intervention groups (*p* = 0.002). This difference was significant between groups E and C (*p* = 0.000) and between groups E and S (*p* = 0.035) after pairwise comparisons; however, according to the results, this was seemingly not the issue. All groups also had comparable comorbidities (hypertension and hyperlipidemia) and were recipients of comparable concomitant medication with beneficial effects on endothelial function (ACEI/ARB or statins); however, both intervention groups had more overall microvascular complications (*p* = 0.046).

The primary endpoint was a change in the arterial function parameters assessed as FMD (stress-induced flow mediated dilatation of brachial artery) and FBF (forearm blood flow as postischemic reactive hyperemia), as well as in parameters of arterial stiffness after 12 weeks of additional treatment with empagliflozin or semaglitide in comparison to control group and between intervention groups. Secondary endpoints were changes in anthropometric measurements and metabolic results.

### 3.2. Results of Anthropometric Measurements

After 12 weeks, body weight (E: −2.49 ± 2.69 kg and S: −4.3 ± 2.98 kg vs. C: −0.10 ± 2.16 kg) and waist (E: −4.0 ± 5.5 cm and −4.4 ± 5.2 cm vs. C: −0–97 ± 5.4 cm) decreased from the baseline in all groups; however, in both intervention groups, the decrement was significant. There was also significant weight reduction in the semaglutide group compared to the empagliflozin group (*p* = 0.020). A slight and nonsignificant decrement was observed in systolic and diastolic blood pressure from baseline to week 12 in both therapeutic groups, along with a minimal increase in heart rate in both therapeutic groups (Table 2 and Table 3).

### 3.3. Results of Glycemia, Lipid Profile, Renal, and Inflamation-Related Issues

The decrement in HbA_1_C from baseline to week 12 was significant in those receiving semaglutide (S: −0.29 ± 0.61 mmol/L, *p* = 0.016), but nonsignificant in those receiving empagliflozin (E: −0.24 ± 0.59 mmol/L, *p* = 0.052); with pairwise comparisons, there was significant decrement in HbA1C in both intervention groups (E–C *p* = 0.015, S–C *p* = 0.003) vs. controls (C: 0.12 ± 0.44, *p* = 0.091). The same applied for prolonged glycemia in the range of 3.9 and 10 mmol/L (TIR—time in range), which was significant in the E group compared to the control (E–C, *p* = 0.07), where, despite the improvement in TIR, decreased hypos were observed. In both therapeutic groups, there was a decrement in glucovariability (glucose excursions expressed as the coefficient of variability), which was non-significantly reduced with both therapeutics. In parallel, the participants in the intervention arms reduced total daily insulin dose (S group, 8.5 IU; E group, 5.1 IU) compared to controls but without between-treatment differences.

The lipid profile was changed, with a small decline in triglycerides and a statistically significant reduction in LDL in the S group (−0.30 ± 0.68 mmol/L, *p* = 0.017), as well as compared to controls (S–C, *p* = 0.044) and the E group (E-S, *p* = 0.001). A reduction in albuminuria (albumin/creatinine ratio) was observed in participants of all groups, mostly in the E group, where there was also a significant increase in cystatine (0.04 ± 0.08, *p* = 0.003) compared to controls (*p* = 0.009) with a slight transient decline in eGFR (calculated as cystatine elimination). There was a significant decline in a marker of inflammation (hs-CRP) in the S group (−0.64 ± 6.4 mmol/L, *p* = 0.024), as well as compared to the other groups (S–C, *p* = 0.015, E–S, *p* = 0.015), and a significant reduction in a risk factor for inflammation (uric acid) after empagliflozin treatment (−22 ± 43 μmol/L, *p* = 0.014). In the E group, there was a significant increase in hematocrit compared to both other groups. Addition of empagliflozin to insulin resulted in only a mild increment in β-hydroxybutyric acid on the control visit (0.02 ± 0.25 mmol/*p* = 0.938). During ketone monitoring at home, there was no alert for ketonemia, and the maximal value recorded during sick days was 1.0 mmol/L (values in DKA were more than 3 mmol/L) (Table 2 and Table 3).

### 3.4. Brachial Diameter/Flow-Mediated Dilation (Change in Brachial Diameter after Ischemia vs. Baseline)

There was a significant improvement in FMD in both therapeutic groups after 12 weeks compared to the baseline (Table 4 and Table 5*).* In the E group, FMD increased from 5.39% ± 3.01% to 10.6% ± 5.7% (2.0-fold, *p* = 0.000); in the S group, it increased from 5.81% ± 3.14% to 11.1% ± 4.9% (1.9-fold, *p* = 0.000); in the control group, we observed the highest FMD at baseline (*p* = 0.024) but without increment after 12 weeks (7.15% ± 2.48% to 7.2% ± 3.2%). There was no difference in responses among therapeutic groups (Figure 1, Table 5).

After adjustment of statistical results for selected confounding covariates (Table 6) in the generalized linear model, we analyzed that statistically significant predictors of better FMD response were treatment with one of the newer antidiabetics (between them, empagliflozin seems to be a stronger predictor) and HbA1C reduction; nonsignificant positive response predictors were a reduction in LDL and the presence of accompanying therapy with ACE-i or statins, while smoking had a nonsignificant negative predicted effect on FMD response. Baseline HbA1C values had no influence on the results (β −0.14, *p* = 0.298); the same applies to changes in glucose variability, changes in TIR, presence of any of microvascular complications, changes in hsCRP or uric acid reduction, body weight change, or duration of diabetes. There was weak positive correlation between hematocrit change and endpoint %FMD2/%FMD1 in the E group (r = 0.172, *p* = 0.107). A reduction in HbA1C seems to be a strong predictor (β = 0.41, *p* = 0.038) of better response. A reduction in HbA1C strongly correlated with endpoint %FMD2/%FMD1 in the E group (r = 0.379, *p* = 0.039), while, in the S group, the same endpoint was not correlated with HbA1C decrement (r = 0.157, *p* = 0.416); we assume that there were differences between both drugs.

### 3.5. Forearm Blood Flow/Reactive Hyperemia (Change in Tissue Perfusion after Ischemia vs. Baseline)

For FBF measurements, the results were variable, with a large number of outliers; thus, only medians were used for interpretation, instead of arithmetic means (Table 4).

At inclusion, baseline FBF and RH in 1 min interval (FBFish − FBFbaz) was not distinguishable among groups. There was a 66% increase in tissue perfusion after cuff release in group E with a median of 1.96 (0.04, 7.7), a 68% increase in group S with a median of 2.08 (0.15, 6.79), and a 60% increase in group C with a median of 2.29 (0.04, 6.42). When comparing the groups at the 3 min interval, there was a non-significantly better flow in the control group: in group E with a median of 0.45 (−0.25, 2.82) (14% increase), group S with a median of 0.54 (−1.0, 3.9) (17% increase), and group C with a median of 0.62 (−2.0, 4.0) (22% increase).

During the study visit, after 12 weeks, slightly higher flow values could be detected already at rest in the empagliflozin group, but these were not significantly higher and there was no statistically significant difference among the groups.

After 12 weeks of therapeutic intervention, there were changes in tissue perfusion that were not significant. The RH in 1 min intervals in group E increased to 97% with a median of 2.55 (0.01, 9.78), while, in group S, it increased to 78% with a median of 3.14 (0.73, 13.0); the proportion increase in this group was even higher because of the low baseline FBF at the control visit, with a median of 2.79 (0.33, 7.86).

At the 3 min interval, tissue perfusion declined toward basal flow; after calculation of mean tissue perfusion per minute (expressed as mL/100 mL tissue/min), the most prolonged postischemic reactive flow was observed in the empagliflozin group.

At the control visit, RH increased within a 3 min interval: in group E with a median of 1.11 (−0.58, 9.9) (35% increase), in group S with a median of 1.08 (−0.62, 13.0) (31% increase), and in group C with a median of 0.85 (−0.90, 6.1) (31% increase). Compared with the baseline RH, the best response with improved tissue perfusion in the first minute was seen in the group treated with empagliflozin with a 1.77-fold increment (or 1.39-fold compared to controls) and in the semaglutide group with a 1.55-fold increment (1.22 fold compared to controls); the lowest response was seen in the controls (1.27-fold increment) *(*Figure 2). In the prolonged interval to 3 min, flow and tissue perfusion declined, but the best perfusion plateau was again reached after empagliflozin treatment (1.68-fold, almost half the response in the other two groups; differences remained nonsignificant among groups).

Baseline HbA1C was negatively correlated only to baseline FBF1 measurements (r = −0.439, *p* = 0.04) with no correlations in postischemic reactive hyperemia. In the generalized linear regression model, there were no significant predictions to better response, changes in glycemic outcomes, or differences in baseline HbA1C (β −0.429, *p* = 0.747), and there were no changes in metabolic outcomes, medical history, or concomitant treatment.

### 3.6. Arterial Stiffness; PWV, Alx, and PR

Measurements in pulse wave velocity did not change in group E (from 7.45 ± 1.47 m/s to 7.50 ± 1.62 m/s, but declined in group S by 4.6% (7.32 ± 1.66 m/s to 6.98 ± 2.0 m/s) and increased in controls to 3.0% (7.32 ± 1.63 m/s to 7.54 ± 1.67 m/s). The augmentation index was also reduced upon receiving semaglutide by 5.5% (29.9 ± 9.1 to 27.4 ± 10.2), whereas it even increased upon receiving empagliflozin or in the controls. Peripheral resistance was diminished in both therapeutic groups; the 5.1% reduction was significant in the semaglutide group (1.74 ± 2.7 mmHg/Lmin decline to 1.65 ± 1.47 mmHg/Lmin) (Table 3 and Table 4, Figure 3). The strong predictors of improved response in the general linear model were semaglutide treatment and systolic blood pressure decline; other important predictors included a lower age and a reduction in LDL (Table 7). Peripheral resistance, PKW, and Alx were also not correlated to any of the intragroup tested variables of glycemic control (∆HbA1C, baseline HbA1C, ∆Glu, ∆TIR, and ∆CV).

### 3.7. Safety and Tolerability

Regarding side-effects, mild side-effects were present; only one drop out was required due to persistent vomiting after semaglutide, while some reported nausea but continued treatment at a reduced dose. In the empagliflozin group, we did not record any side-effects. During the period between the two study visits, participants did not experience a significant increase in ketone bodies—beta-hydroxybutyrate—up to a maximum of 1.0 mmol/L during the cold period. During the control visit, we recorded a minimal increase from baseline (Table 3), but this was statistically insignificant (*p* = 0.938). This suggests a safer, reduced dose profile in insulinopenic patients.

## 4. Discussion

The present study ENDIS revealed that a 12 week course of empagliflozin or semaglutide improved endothelium function in the macro- and microcirculation of patients with T1DM without coexisting cardiovascular or renal diseases when compared to controls.

Improvement in FMD was significant in both intervention groups compared to controls with no changes between the two groups. In arterial stiffness parameters, improvements were seen only in the semaglutide group, with a significant decline in peripheral resistance. During the evaluation of FBF, which is a method for microvascular endothelium function assessment, there were statistically insignificant improvements in both therapeutic groups compared to controls.

In our study, we tried to achieve as comparable groups as possible with respect to glycemic and other metabolic factors, as well as factors that already have a proven impact on endothelial function. Unfortunately, the empagliflozin-treated group and the control group appeared to differ significantly in baseline HbA1C; however, after adjustment, this difference did not prove to be significant for the final observed events.

A significant reduction in HbA1C seemed to have the only strongly predictive relationship with FMD-related outcome, but this is more related to empagliflozin, while the semaglutide effect is independent of this. The outcomes were also not affected by the otherwise nonsignificant reduction in glucovariability in both treatment groups, or by the incidence of hypoglycemia, which was very low in all groups. Despite the significant weight loss and reduced waist circumference, indicating a reduction in visceral fat in both treatment groups, this also did not contribute to beneficial effects, which were more favorable in group E. The same can be said for the statistically nonsignificant reduction in both systolic and diastolic blood pressure, as well as triglycerides, and the significant reduction in LDL cholesterol and hsCRP (both seen only in group S). We can exclude the impact of the statistically significantly decrease in uric acid in group E, Lastly, regarding our study results, we can conclude with high confidence that the beneficial effects of empagliflozin on macrovascular endothelial function are at least partly related to HbA1C reduction, while semaglutide effects are more of a reflection of the pleiotropic effect of the drug itself on NO-related dilation, and not simply an effect on glycemic or other metabolic and cardiovascular indices.

The results of the second part of observed macrovascular function (arterial stiffness) were less significant. However, we observed a reduction in all indicators of arterial stiffness and even a statistically significant reduction in peripheral resistance in those receiving semaglutide. In our study in group E, we observed virtually no difference after the introduction of the drug. The only strong predictor of arterial stiffness improvement was a reduction in blood pressure; positive predictors also included LDL reduction and younger age, while glycemic and CRP changes did not have any predictive value.

Although we did not achieve statistical power in the evaluation of reactive stress-induced microvascular hyperemia, we can conclude that there is a beneficial effect, which is particularly pronounced with empagliflozin treatment. Empagliflozin already had an effect on the flow increase during the control visit under basal conditions. The flow as a consequence of reactive postischemic hyperemia at the control visit compared to the reactive postischemic flow at the baseline visit was also higher, compared to the control group. Not only was the maximal response increased, but, in this group, a prolonged response with prolonged hyperemia was observed even during the 3 min period after cuff release. A slightly lower increase in flow was observed in the semaglutide group, but there was no significant difference compared to the control group. However, compared to empagliflozin, flow was as much as twofold lower throughout the 3 min postischemic period.

The only published study on endothelium dysfunction research comparing SGLT 2 and incretin therapy revealed that low-mediated dilation significantly increased after the 3 month treatment period in the empagliflozin group (4.8% ± 4.5% vs. 8.5% ± 5.6%, *p* = 0.03), while no change was detected in the incretin-based therapy group (5.1% ± 4.5% vs. 4.7% ± 4.7%, *p* = not significant) [32].

Among the publications, the most comparable to our study was performed in subjects with T1DM by Lunder et al., showing a statistically significant response in endothelial function assessed by FMD in all intervention groups, with the greatest response seen in the group treated with a combination of empagliflozin and metformin (up to 2.6-fold, *p* < 0.001), FMD significantly improved after 12 weeks of treatment with empaglifozin (up to 2.2-fold,) but it was also the only group with a statistically significant reduction in HbA1C; adjustments to glycemic outcomes were not performed in the previous study [27].

Considering the studies published with the effect of SGLT 2 inhibitors on arterial stiffness, the results are controversial. Pooled meta-analysis of five studies showed no significant differences in the change in PWV between the SGLT-2i group and the control group (SMD: 0.11, 95% CI: −0.15–0.37, *p* = 0.4) [33]. Since the nonsignificant results were generally in studies with dapagliflozin, experts have formed the opinion [34] that there is a difference between dapagliflozin and empagliflozin, at least with regard to the effect on arterial stiffness and its relevance in long-term cardiovascular studies. This is why we expected more favorable results in group E, as was also seen in a study with a comparative selection of participants who showed an improvement in arterial stiffness [27]. When compared to placebo, empaglifozin or empaglifozin/metformin significantly improved PWV (up to 5.1- and 5.7-fold, both *p* < 0.01). The uncharacteristic values for arterial stiffness parameters in our study may have been due to the use of a lower dose of empagliflozin than in some comparable studies [25,27]. In another study, which compared empagliflozin to placebo, the significant reduction in hsCRP was the only predictor of a reduction in arterial stiffness, in addition to a reduction in blood pressure [24]. We can compare this with our study, where a strong predictor for arterial stiffness improvement was also a reduction in blood pressure, while a significant reduction in hsCRP was achieved in the semaglutide group, albeit without predictive value. In the absence of comparative studies on the effect of semaglutide on this parameter, parallels can be drawn with published studies in T2DM for other GLP 1 agonist agents, which showed significant reductions in PWV (two studies; *n* = 62 patients, pooled MD = −0.18, 95% CI: −0.30 to −0.07, *p* = 0 002) associated with a significant reduction in PWV (pooled MD = −1.97, 95% CI: −2.65 to −1.30, *p* < 0.001) [26]. In a study investigating the effects of 6 month treatment with the glucagon-like peptide-1 analogue liraglutide on arterial stiffness, left-ventricular myocardial deformation, and oxidative stress in subjects with newly diagnosed type 2 diabetes [32], the groups were equally randomized with respect to all entry parameters, both anthropometric and metabolic, as well as smoking status, ACE I, and statin intake [30]. However, markers with an impact on the inflammatory response, e.g., LDL cholesterol or hsCRP, were not determined. After adjusting for HbA1c, weight, BMI, and waist circumference, patients that received liraglutide had lower PWV, AI, systolic blood pressure, and central systolic blood pressure than those on metformin (*p* < 0.05). Another cited study [34], which was not controlled, examined the effects of short-term exenatide treatment on the aortic pulse wave velocity of obese T2DM patients, confirming a decline in aortic PWV from 7.2 ± 2.2 m/s at baseline to 5.1 ± 0.1 m/s after treatment (*p* = 0.001); furthermore, there was a significant association with a reduction in waist and hsCRP, but only in younger group of participants. When comparing the results of this and our study, a better predictor for arterial stiffness improvement is younger age; hence, we can speculate that younger patients exhibit a more beneficial effect of GLP 1 agonists on arterial stiffness. However, the time of the study was likely too short to draw any major conclusions regarding this matter.

According to a microvascular endothelial assessment in a similar study conducted on T1DM, reactive hyperemia index (RHI) significantly improved with empaglifozin (up to 1.3-fold, *p* < 0.01) [27]. We could speak of a comparable effect of empagliflozin in our study, although it did not reach statistical power. The methods of microvascular endothelial function assessment were not the same. In the publications of other SGLT 2 inhibitors, no significant effect on reactive hyperemia and microcirculation was observed. There are no comparable studies performed with GLP 1agonists on microvascular endothelial assessment. There have been some promising studies in the past on the effect of GLP 1 agonists on this type of vascular network, particularly using native GLP 1 [34]. There were conclusions that endothelial function can potentially be improved by reducing glucose fluctuations by decreasing postprandial hyperglycemia, by reducing postprandial triglyceride levels, and by activating AMPK (AMP-activated protein kinase) [35].

Lastly, it should be reiterated that each antidiabetic drug class can be associated with differences in the vascular effects. Irrespective of the different correlations, there is certainly synergy of the individual metabolic effects on different levels of the vascular network and their vascular response. On the basis of several lines of evidence, we can conclude that the effect of lowering blood pressure plays an important role in arterial stiffness and peripheral resistance, and that the effect on the lipid profile takes precedence over the effect of lowering glycemia, which seems to be more important in ensuring the function of the endothelium of large vessels. Even fewer conclusions can be drawn at the microcirculatory level, where none of the changes in metabolic effects appear to have predictive value for improving reactive hyperemia function. However, the differences in metabolic outcomes are so small; therefore, larger groups of participants would be needed to provide answers. In addition to glycemic reduction, the reduction in glucose variability plays an important role, although we could not confirm its importance in our study. Nevertheless, we have to be aware that the population of patients included was extremely well-controlled, and, with the use of the CGM system, their glucose variability was also low and not entirely comparable to most of the less well-managed diabetic patients we encounter in our daily practice. Experimental studies have indicated the possibility that a dysregulated autoimmune response in T1DM may contribute to endothelial dysfunction by increasing oxidative stress through activation of NADPH oxidase [36]. It still remains unclear whether glucose fluctuations are involved in endothelial dysfunction in patients with type 1 diabetes. All these facts should lead us to strive for better regulation and minimized glucose excursion, along with the use of advanced technologies for the treatment of type 1 diabetes.

New antidiabetic drugs empagliflozin and semaglutide are associated with metabolic, cardiovascular, and many pleiotropic effects, thus also being called diabetes-modulating drugs, because they alter the course and prognosis of the disease. The complexity in understanding endothelial function and the influence of new drugs on it are not completely known in T1DM; our study gives us some new answers and perspectives.

## 5. Conclusions

Through the ENDIS study, we have added another piece to the mosaic of understanding the full pathophysiological mechanisms of the newer antidiabetic drugs; for the first time, a comparison of not only metabolic but also vascular effects was conducted between empagliflozin and semaglutide. We can conclude that both empagliflozin and semaglutide have a positive impact on glycemic control and other metabolic outcomes in T1DM. For arterial stiffness, semaglutide seems better then empagliflozin at a low dose, but both improve endothelial function. Regardless of the area of the vascular bed, the effects are more pronounced in macro- than microvascularization; thus, these drugs have a protective role, contributing to a slower reduction in cardiovascular risk. We have to weigh the potential short- and long-term benefits of these adjunct therapies versus the potential acute side-effects as it is clear that the overall risk for DKA in patients taking empagliflozin remains very low due to proper education of them and their caretakers. Overall, this makes the use of these drugs in T1D worth considering.

## Figures and Tables

**Figure 1 pharmaceutics-15-01945-f001:**
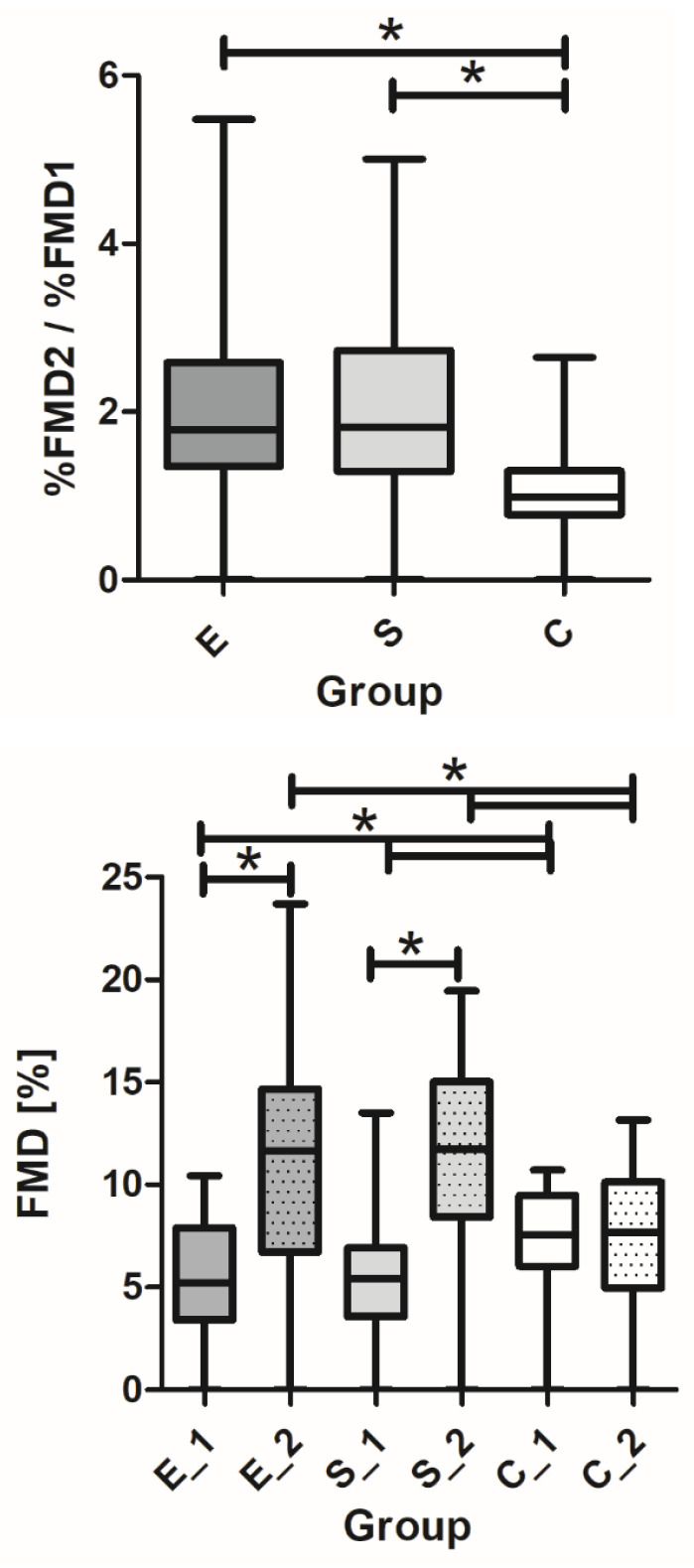
Baseline FMD and increments after 12 weeks in three groups: E—empagliflozin, S—semaglutide, and C—control (**bottom**); ratio of FMD between two visits (**top**). * *p* < 0.05 significance level.

**Figure 2 pharmaceutics-15-01945-f002:**
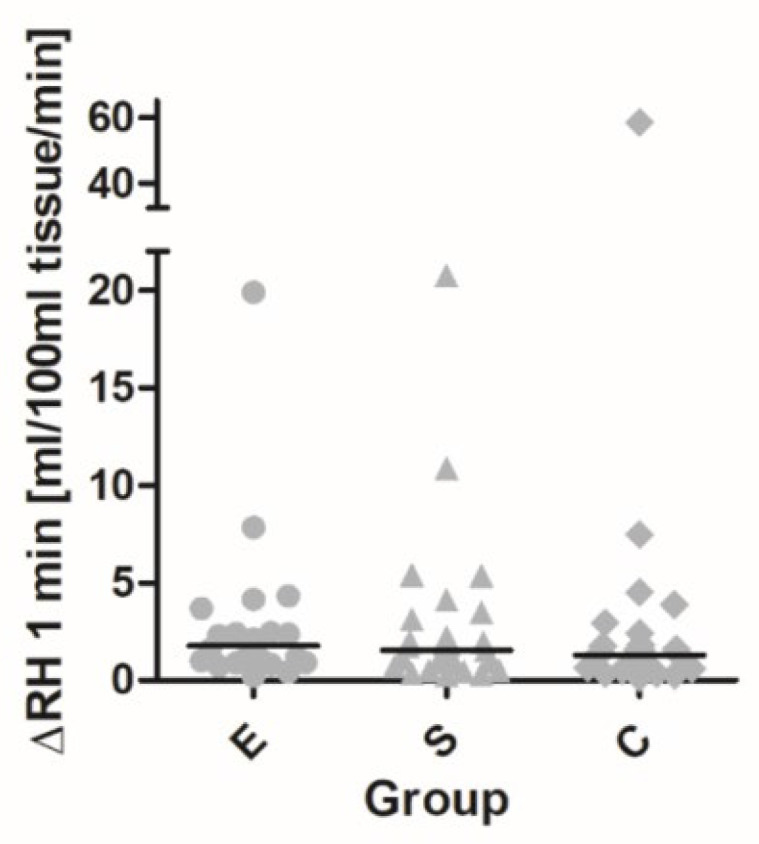
Ratio in tissue perfusion in reactive postischemic hyperemia (1 min interval) between control and first visit in two therapeutic groups (E—empagliflozin and S—semaglutide) and one control group (C).

**Figure 3 pharmaceutics-15-01945-f003:**
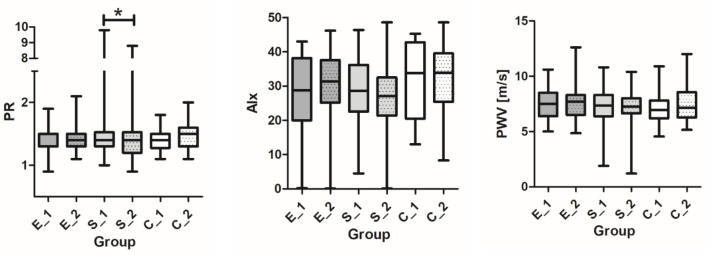
Changes in peripheral resistance (PR), augmentation index (Aix75), and pulse wave velocity (PWV) before and after 12 weeks in two therapeutic groups (E—empagliflozin and S—semaglutide) and one control group (C). * *p* < 0.05 significance level.

**Table 1 pharmaceutics-15-01945-t001:** Participant disposition, demographics, and baseline characteristics.

Baseline Caracteristics	Empagliflozin (*n* = 30)	Semaglutide (*n* = 30)	Control (*n* = 29)	*p*
Age (years)	48.2 ± 10.7	48.5 ± 9.6	47.0 ± 12	0.697
DM duration(years)	21.3 ± 10.2	21.9 ± 11.2	18.8 ± 11.9	0.548
Sex male (*n*/%)	18/60	19/63.3	17/58.6	0.930
BMI (kg/m^2^)	27.7 ± 4.3	28.2 ± 4.8	26.9 ± 3.0	0.783
Weight (kg)	81.9 ± 14.2	84.6 ± 16.4	82.6 ± 11.7	
Waist (cm)	99.9 ± 11.7	99.5 ± 14.3	98.5 ± 9.9	
Smokers (*n*/%)	4/13.3	5/16.7	5/17.2	
HbA1C (%)	7.88 ± 0.72	7.42 ± 0.80	7.04 ± 0.90	0.002 *
Hypos (%)	1.9 ± 2.8	4.2 ± 4.3	3.3 ± 3.1	
Glucose (mmol/L)	9.02 ± 2.93	7.62 ± 2.84	7.55 ± 3.04	
TIR > 70% (*n*/%)	10/33.3	19/63.3	17/58.6	
TIR (%)	61.7 ± 17.1	70.6 ± 14.7	69.7 ± 17.4	
CV ≤ 36% (*n*/%)	22/76.7	29/55.2	19/67.9	
CV (%)	33.8 ± 5.7	35.3 ± 6.1	33.9 ± 5.5	
MDI basal–bolus (*n*)	13	17	17	
CSII without loop (*n*)	3	4	3	
CSII loop (*n*)	4	2	2	
CSII Hybride (*n*)	10	7	7	
Hypertension (*n*/%)	15/50	15/50	12/41.4	
RR sist (mmHg)	132.5 ± 15.7	134.7 ± 16.4	130.5 ± 13.6	0.875
RR diast (mmHg)	76.5 ± 7.4	78.4 ± 15.5	75.7 ± 7.5	0.979
Heart rate (/min)	66.4 ± 10.5	68.3 ± 9.9	68.6 ± 12.6	
Hyperlipidemia (*n*/%)	22/73.3	22/73.3	19/65.5	
LDL (mmol/L)	2.61 ± 0.86	2.66 ± 0.71	2.62 ± 0.78	0.948
HDL (mmol/L)	1.56 ± 0.36	1.53 ± 0.38	1.51 ± 0.39	0.830
Triglycerides (mmol/L)	1.07 ± 0.51	1.23 ± 0.75	1.11 ± 0.56	0.900
eGFRcys (mL/min/1.73)	86.1 ± 9.2	85.8 ± 8.3	86.5 ± 7.8	0.872
ACR (g/mol)	3.35 ± 11.2	2.41 ± 2.92	2.35 ± 5.54	
All MVC (*n*/%)	14/46.7	19/63.3	9/31	0.046 *
NPDR (*n*/%)	8/26.6	11/36.6	5/16.6	
D. neuropathy (*n*/%)	3/10	3/10	3/10	
MAU (*n*/%)	6/20	10/33.3	5/17.2	
CM (*n*/%)	17/43.3	13/43.3	9/31	0.139
ACE inh/ARB (*n*)	14	9	7	
Statin (*n*)	13	9	6	
hs CRP (mmol/L)	2.32 ± 2.1	5.5 ± 8.2	2.5 ± 3.1	
UA (μmol/L)	252 ± 72	245 ± 62	252 ± 61	

BMI—body mass index, Ht—hematocrit, TIR—time in range (Glu 3.9–10 mmol/L), CV—coefficient of variability (glucose excursions), MDI—multiple daily injections, CSSI—continuous subcutaneous insulin infusion ACR—albumin/creatinine ratio, MVC—microvascular complications (summary), NPDR—non-proliferative diabetic retinopathy, MAU—microalbuminuria, CM—concomitant medications (summary of ACE/ARB and statin recipients), UA—uric acid. * *p* < 0.05 significance level.

**Table 2 pharmaceutics-15-01945-t002:** Secondary endpoints: anthropometric, metabolic, renal, and inflammation endpoints.

Variables	Empagliflozin (*n* = 30)	Semaglutide (*n* = 30)	Control (*n* = 29)	*p ***
	Mean ± SD Median (Min, Max)	Mean ± SD Median (Min, Max)	Mean ± SD Median (Min, Max)	
∆BMI (kg/m^2^)	−0.88 ± 1.07 −0.85 (−3.5, 0.8) *p* = 0.000 *	−1.42 ± 4.8 −1.5 (−3.8, 0.69) *p* = 0.000	0.04 ± 0.98 −0.30 (2.4, 1.9) *p* = 0.782	0.000 ****
∆Weight (kg)	−2.49 ± 2.69 −2.8 (−10.0, 1.0) *p* = 0.000 ***	−4.3 ± 2.98 −4.2 (−11.7, 1.0) *p* = 0.000 ***	−0.10 ± 2.16 −1.0 (−3.7, 4.0) *p* = 0.791	0.000 ****
∆Waist (cm)	−4.0 ± 5.5 3.5 (−19, 9) *p* = 0.000 ***	−4.4 ± 5.2 −3.5 (−24, 2) *p* = 0.000 ***	−0–97 ± 5.4 0.0 (−14, 12) *p* = 0.395	0.016 ****
∆HbA1C (%)	−0.24 ± 0.59 −0.1 (−1.8, 0.9) *p* = 0.052	−0.29 ± 0.61 −0.39 (−2.0, 0.9) *p* = 0.016 ***	0.12 ± 0.44 0.1 (−1.1, 1.1) *p* = 0.091	0.00 ****
∆TIR (%)	4.2 ± 11.9 6.0 (−25, 28) *p* = 0.393	0.7 ± 13.7 0.5 (−53, 20) *p* = 0.247	−1.35 ± 8.8 −2.5 (−15, 19) *p* = 0.230	0.027 ****
∆CV (%)	−0.91 ± 5.0 −1.5 (−13.0, 10.3) *p* = 0.221	−1.28 ± 4.8 −1.2 (−12, 11.4) *p* = 0.143	1.06 ± 4.3 0.7 (−6.4, 16.2) *p* = 0.223	0.103
∆hypo (%)	−0.71 ± 2.3 0.0 (−9, 2.5) *p* = 0.176	0.22 ± 4.4 0.0 (−4.6, 6.3) *p* = 0.858	−0.48 ± 2.3 0.0 (−6.9, 4.0) *p* = 0.443	0.180
∆TDI (IU)	−5.1 ± 10.0 −2.7 (−31, 13) *p* = 0.009 ***	−8.5 ± 9.3 −6.0 (−32, 7) *p* = 0.000 ***	−1.1 ± 7.5 0.0 (−32, 10) *p* = 0.747	0.001 ****
∆RR syst (mmHg)	−1.7 ± 19 −4.0 (−31, 55) *p* = 0.271	−3.7 ± 18 −3 (−42, 29) *p* = 0.410	5.4 ± 16 8 (−20, 53) *p* = 0.138	0.105
∆RR diast (mmHg)	−1.8 ± 7.5 −2.5 (−22, 15) *p* = 0.174	−3.0 ± 14 −1.0 (−43, 28) *p* = 0.381	2.8 ± 7.8 3.0 (−12, 16) *p* = 0.067	0.066
∆heart rate (/min)	1.5 ± 9.0 −1.0 (−15, 24) *p* = 0.681	1.7 ± 6.6 2.5 (−12, 12) *p* = 0.115	−0.6 ± 7.7 0.0 (−21, 12) *p* = 1.0	0.512
∆LDL (mmol/L)	0.03 ± 0.75 0.00 (−2.8, 1.4) *p* = 0.393	−0.30 ± 0.68 −0.5 (−1.2, 1.9) *p* = 0.017 *	−0.15 ± 0.51 −0.1 (−1.6, 0.9) *p* = 0.129	0.003 ****
∆HDL (mmol/L)	0.003 ± 0.26 0.0 (−0.9, 0.6) *p* = 0.741	−0.05 ± 0.25 0.0 (−0.5, 0.6) *p* = 0.186	0.02 ± 0.25 0.0 (−0.5, 0.6) *p* = 0.723	0.439
∆Tg (mmol/L)	−0.01 ± 0.39 0.0 (−1.1, 0.9) *p* = 0.978	−0.12 ± 0.44 −0.1 (−1.4, 0.7) *p* = 0.178	0.09 ± 0.53 0.0 (−0.8, 2.1) *p* = 0.556	0.346
∆eGFR (cystatine)	−1.67 ± 5.0 0.0 (−20, 8) *p* = 0.075	0.83 ± 4.2 0.0 (−13, 13) *p* = 0.202	−0.62 ± 2.3 0.0 (−9, 3) *p* = 0.234	0.075
∆cystatine (mg/L)	0.04 ± 0.08 0.03 ± (0.15, 0.35) *p* = 0.003 ***	−0.003 ± 0.08 0.0 (−0.1, 0.25) *p* = 0.547	−0.26 ± 1.48 0.08 (−8, 0.1) *p* = 0.516	0.025 ****
∆ACR (mg/L)	−1.08 ± 5.0 0.0 (−27, 1.2) *p* = 0.467	−0.26 ± 2.56 −0.49 (−7.8, 8.3) *p* = 0.124	−0.62 ± 3.0 0.0 (−15.8, 2.1) *p* = 0.273	0.639
∆hsCRP (mmol/L)	1.69 ± 7.4 0.15 (−3.5, 40) *p* = 0.234	−0.64 ± 6.4 −0.25 (−10.6, 21) *p* = 0.024 *	0.73 ± 4.6 0.1 (−10.6, 21) *p* = 0.247	0.019 ****
∆UA (μmol/L)	−22 ± 43 −22 (−112, 58) *p* = 0.014 ***	0.47 ±45 0.5 (−100, 111) *p* = 0.880	0.55 ± 32 5.0 (−84, 75) *p* = 0.634	0.065
	0.02 ± 0.25			
∆βOHB	0.0 (−0.5, 0.9)			
(mmol/L)	*p* = 0.938			
∆Ht	0.468 ± 2.4	−0.017 ± 0.07	−0.001 ± 0.025	0.005 ****
	*p* = 0.008 ***	*p* = 0.225	*p* = 0.538	

BMI—body mass index, TIR—time in range (3.9 to 10 mmol/L), CV—coefficient of variability of glucose excursions, TDI—total daily insulin dose, Tg—triglycerides, ACR—albumin to creatinine ratio, hsCRP—high-sensitivity C reactive protein, UA—uric acid, βOHB—hydroxybutyric acid, Ht—hematocrit. * *p* < 0.05 significance level, **** independent-samples Kruskal–Wallis test *p* < 0.05 significance level.

**Table 3 pharmaceutics-15-01945-t003:** Pairwise comparisons for significant results of secondary endpoints.

Variable	E–C *p*	S–C *p*	E–S *p*
∆BMI (kg/m^2^)	0.006 *	0.000 *	0.075
∆Weight (kg)	0.006 *	0.000 *	0.020 *
∆Waist (cm)	0.016 *	0.010 *	0.857
∆HbA1C (%)	0.015 *	0.003 *	0.547
∆TIR (%)	0.007 *	0.124	0.252
∆TDI (IU)	0.022 *	0.000 *	0.114
∆LDL (mmol/L)	0.192	0.044 *	0.001 *
∆cystatine (mg/L)	0.009 *	0.057	0.491
∆hsCRP (mmol/L) ∆Ht	0.996 0.012 *	0.015 * 0.596	0.015 * 0.002 *

* *p* < 0.05 significance level; significance values were adjusted by Bonferroni correction for multiple tests.

**Table 4 pharmaceutics-15-01945-t004:** Primary endpoint: endothelial function assessment.

Variables	Empagliflozin (*n* = 30)	Semaglutide (*n* = 30)	Control (*n* = 29)	*p*
**FMD** **Measurements**	**Mean ± SD** **Median (Min, Max)** ** *p* **	**Mean ± SD** **Median (Min, Max)** ** *p* **	**Mean ± SD** **Median (Min, Max)** ** *p* **	
FMD1 (mm)	0.20 ± 0.11 0.21 (0.00, 0.4) *p* = 0.000 *	0.23 ± 0.11 0.23 (0.00, 0.5) *p* = 0.000 *	0.27 ± 0.09 0.26 (0.00, 0.40) *p* = 0.000 *	0.024 **
% FMD1 (%)	5.39 ± 3.01 5.21 (0.00, 10.4)	5.81 ± 3.14 5.45 (0.00, 13.5)	7.15 ± 2.48 7.54 (0.00, 10.7)	
FMD2 (mm)	0.41 ± 0.22 0.45 (0.22, 1.06) *p* = 0.000 *	0.44 ± 0.18 0.40 (0.0, 0.7) *p* = 0.000 *	0.28 ± 0.13 0.30 (0.00, 0.56) *p* = 0.000 *	0.001 **
% FMD2 (%)	10.6 ± 5.7 11.6 (0.00, 23.6)	11.1 ± 4.9 11.7 (0.3, 19)	7.2 ± 3.2 7.7 (0.00, 13)	
FMD2 − FMD1 (mm)	0.209 ± 0.18	0.211 ± 0.14	0.008 ± 0.11	
	0.233 (−0.1, 0.9)	0.217 (0.0, 0.5)	0.00 (−0.3, 0.2)	0.000 **
%FMD2/%FMD1	2.0 ± 1.3	1.9 ± 1.3	1.0 ± 0.5	
	1.7 (0.0, 5.4)	1.7 (0.0, 5.0)	0.9 (0.0, 2.6)	0.000 **
	*p* = 0.000 *	*p* = 0.000 *	*p* = 0.427	
**FBF measurements**	**Median (Min, Max)** ** *p* **	**Median (Min, Max)** ** *p* **	**Median (Min, Max)** ** *p* **	
Baseline FBF1	2.87 (1.3,6.3)	3.58 (1.33, 9.0)	3.2 (1.44, 7.5)	0.418
RH 1 (1 min interval) Ml/100 mL/min/%	1.96 (0.04, 7.7) =66%	2.08 (0.15, 6.79) =68%	2.29 (0.04, 6.42) =60%	0.684
RH 1 (3 min interval) Ml/100 mL/min/%	0.45 (−0.25, 2.82) =14%	0.54 (−1.0, 3.9) =17%	0.62 (−2.0, 4.0) =22%	0.685
Baseline FBF2	3.47 (0.7, 5.9)	3.25 (0.5, 5.2)	2.82 (0.5, 5.2)	0.327
RH 2 (1 min interval) Ml/100 mL/min	2.55 (0.01, 9.78) =97% *p* = 0.074	3.14 (0.73, 13.0) =78% *p* = 0.701	2.79 (0.33, 7.86) =103% *p* = 0.990	0.912
RH 2 (3 min interval) Ml/100 mL/min/%	1.11 (−0.58, 9.9) =35% *p* = 0.101	1.08 (−0.62, 13.0) =31% *p* = 0.564	0.85 (−0.90, 6.1) =31% *p* = 0.675	0.973
RH2/RH1 1 min	1.77 (0.21, 19.8)	1.55 (0.26, 20.7)	1.2 (0.14, 58.4)	0.552
RH2/RH1 3 min	1.68 (−52, 35)	0.72 (−2.4, 24.8)	0.71 (−13, 8.0)	0.445
**Arterial stiffness**	**Mean ± SD** **Median (Min, Max)** ** *p* **	**Mean ± SD** **Median (Min, Max)** ** *p* **	**Mean ± SD** **Median (Min, Max)** ** *p* **	
∆ PWV (m/s)	0.04 ± 0.82 0.0 (−2.0–2.0) *p* = 0.923	−0.29 ± 1.43 0.0 (−7.3–1.1) *p* = 0.483	0.27 ± 0.82 0.2 (−0.8–3.4) *p* = 0.109	0.250
∆ Alx 75 (%)	2.55 ± 8.7 2.0 (−12, 28) *p* = 0.149	−1.42 ± 8.0 −0.5 (−19, 18) *p* = 0.248	1.56 ± 7.8 0.00 (−9.6, 27) *p* = 0.501	0.209
∆ PR (mmHg/Lmin)	−0.01 ± 0.17 0.0 (−0.4–0.4) *p* = 0.754	−0.07 ± 0.23 0.0 (−1.0–0.3) *p* = 0.046 *	0.06 ± 0.06 0.37 (0.2–0.5) *p* = 0.060	0.025 **

FMD—flow-mediated vasodilatation of brachial artery, FBF—forearm blood flow, RH—reactive hyperemia, PWV—pulse wave velocity, PR—peripheral resistance, Alx75—augmentation index adjusted to heart rate 75/min, BMD—brachial artery diameter, FMD (mm) = BMDish – BMDbaz, FMD (%) = (BMDish – BMDbaz)/BMDbaz, RH (mL/100 mL/min) = FBFish − FBFbaz, RH (%) = (FBFish − FBFbaz)/FBFbaz. * *p* < 0.05 significance level, ** independent-samples Kruskal–Wallis test *p* < 0.05 significance level.

**Table 5 pharmaceutics-15-01945-t005:** Pairwise comparisons for significant results of primary endpoints.

Variable	E–C *p*	S–C *p*	E–S *p*
FMD 1	0.035 *	0.088	1.0
%FMD1	0.015 *	0.029 *	0.810
FMD 2	0.013 *	0.001 *	1.0
%FMD2 FMD2-FMD1	0.004 * 0.000 *	0.000 * 0.000 *	0.461 0.709
%FMD2/%FMD1	0.000 *	0.000 *	0.745
∆ PR	0.119	0.007 *	0.258

* *p* < 0.05 significance level; significance values were adjusted by Bonferroni correction for multiple tests.

**Table 6 pharmaceutics-15-01945-t006:** Predictors of better response on reactive hyperemic flow (FMD).

	β	SE	CI95 L	CI95 U	*p*
Intercept	0.800	0.2627	0.285	1.314	0.002
Smoking	−0.265	0.2744	−0.803	0.272	0.333
AH	0.098	0.3051	−0.500	0.696	0.748
HLP	0.064	0.2630	−0.451	0.580	0.807
MVC	−0.001	0.2563	−0.504	0.501	0.996
CM	0.558	0.3122	−0.054	1.179	0.074
Duration	−0.001	0.0107	−0.022	0.020	0.904
∆CV	0.024	0.0230	−0.021	0.069	0.300
∆BW	−0.060	0.0401	−0.138	0.019	0.136
∆TIRp	0.008	0.0097	−0.011	0.027	0.432
∆HbA1C	0.410	0.1976	−0.022	0.797	0.038 *
∆LDL ∆CRP	0.306 0.009	0.1677 0.0172	−0.022 −0.025	0.635 0.043	0.068 0.602
∆UA	0.001	0.0025	−0.004	0.006	0.663
Group E	0.793	0.2938	0.217	1.368	0.007 *
Group S	0.820	0.3258	0.182	1.459	0.012 *

Covariates appearing in the model were fixed at the following values: duration = 20.69; ∆_TT = −2.262; ∆_CV = −0.375; ∆_TIRp = 1.132; ∆_HbA1C = −0.137; ∆_LDL = −0.134; ∆_urati = −7.79; ∆_CRP = 0.613. HbA1C (baseline) β −0.14 *p* = 0.298 was fixed in another model as it could not appear in the same model as ∆HbA1C. * *p* < 0.05 significance level.

**Table 7 pharmaceutics-15-01945-t007:** Predictors of improved response in peripheral resistance (PR).

Variable	β	SE	CI95 L	CI95 U	*p*
Intercept	0.200	0.1018	0.001	0.400	0.049
Smoking	−0.027	0.0556	−0.136	0.082	0.627
AH	0.001	0.0485	−0.094	0.096	0.980
HLP	0.080	0.0501	−0.018	0.179	0.108
Age	−0.004	0.0023	−0.009	0.059	0.052
∆RRsyst	0.003	0.0012	−0.001	0.005	0.0.13 *
∆LDL	−0.064	0.0335	−0.130	0.001	0.055
∆HbA1C	0.020	0.0335	−0.049	0.090	0.565
Group E	−0.033	0.0492	−0.129	0.063	0.502
Group S	−0.114	0.0494	−0.017	5.352	0.021 *

AH—arterial hypertension, HLP—hyperlipidemia, RRsyst—systolic blood pressure. Covariates appearing in the model were fixed at the following values: age = 48.09; ∆_HbA1C = −0.140; ∆_LDL = −0.131; ∆_sRR = 0.17; ∆_CRP = 0.611; HbA1C (baseline) β −0.14 *p* = 0.298 was fixed in another model as it could not appear in the same model as ∆HbA1C. * *p* < 0.05 significance level.

## Data Availability

The data supporting the findings of this study are available from the corresponding author upon reasonable request (maja.navodnik-preloznik@sb-celje.si).

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
