# Peer review of "The Effect of Additional Treatment with Empagliflozin or Semaglutide on Endothelial Function and Arterial Stiffness in Subjects with Type 1 Diabetes Mellitus—ENDIS Study"

_pharmaceutics, 2023, doi:10.3390/pharmaceutics15071945_

Round 1
Reviewer 1 Report
The title of the manuscript is good. English language has good quality. Tables need some adjustments. Some sections of the manuscript need some changes. There are some modifications that need to be exerted in the citations.
1. Please reform the section "Abstract" based on order below:
Write briefly about:
+ the importance of treatmen of diabetes mellitus
+ the importance of your work
+ material and method
+ results
+ conclusions
2. There are some abbreviations in the text of manuscript, for example GLP-1, T1DM and etc.. The authors should define alk of them when thay use them for the first time.
3. Line 53 in page 2
This line does not have proper link with its previous paragraph. Please create this link in order to keep the continuity of the text.
4. Line 62-68 in page 2
Why this part contains no reference?
5. All multipple and middle sentence references in all over the manuscript should be reformed.
6. Line 72-75 in page 2
This part has no reference. Why?
7. About the part "2.1. Study design and subject selection" in line 84
Do the authors perform this part based on previous studies? If yes, which study? And why the have not mentioned the name of this survey?
If no, please explain why you performed this
section in this style?
8. Line 199-201 in page 5
This part does not belong to the section "Results". It belongs to the part "material and methods". Please reconsider it.
9. Why the format of Table 1 is different from Table 2? The format of all tables should be the same and according to the guidline of journal.
10. About Table 1
Please reform the title of this table. Its vague and does not have enough information to introduce this table to other readers.
12. line 308 in page 13
Please reform the name "Table 5". It should be "Table 6"
13. Why the format of tables in this
manuscript is different from each other? Please reform the format of all tables based on the guidline of journal.
14. About the part "Discussion"
Please rewrite this part according to notes below:
First: categorize all of your results based on their importance (from the most important one to the least important)
Second: after that, turn each one of your results into some subheadings
Third: after that, discuss about them one by one
Forth: make comparisons between your results and the results of other similar and relevant surveys
15. This is important that the authors compare their findings with the results of prior similar surveys in the part "Discussion".
16. Please check and adjust the "Reference list" based on the regulations of reference list of journal. (Titles, doi, the name of journal and ... )
Author Response
Response to Reviewer 1 Comments
Point 1. Please reform the section "Abstract" based on order below:
Write briefly about:
+ the importance of treatmen of diabetes mellitus
+ the importance of your work
+ material and method
+ results
+ conclusions
Response 1:
In the instructions for preparing the Abstract for this Journal, it is explicitly stated that it should not contain subtitles and is limited to maximum of 200 words, which was used for presentation of the study design, methods results and conclusions and there were no place to write about importance of treatment of DM or importance of my work.
Point 2. . There are some abbreviations in the text of manuscript, for example GLP-1, T1DM and etc.. The authors should define alk of them when thay use them for the first time.
Response 2:
We can add explanation of abbreviation for both drugs : GLP 1 (glucagon-like peptide 1) agonists and SGLT 2 (sodium – glucose cotransporter -2 ) inhibitors. All other abbreviations was explainted in the manuscript when appeared for the first time ( T1DM, CGMS, FMD, PWV, FBF etc)
Point 3. Line 53 in page 2
This line does not have proper link with its previous paragraph. Please create this link in order to keep the continuity of the text.
Response 3:
The sentences before and after in line 53 can be created in changed order to keep continuity from previous paragraph. Corrections of reference numbering will be done later if you agree with changes.
- changed previous lines from 34 to 56
Multifactorial approach beyond the glucose-lowering effect of these drugs is crucial in routine clinical practice to reduce cardiovascular risk in individuals with T2DM. Besides T2DM, T1DM is also associated with higher mortality and cardiovascular disease (CVD) risk than the general population [5,6]. Hyperglycemia appears to have a more profound effect on cardiovascular risk in T1DM while in T2DM achievement of treatment targets for all recognized risk factors is crucial [7,8]. A pivotal role in cardiovascular health and disease holds the vascular endothelium, which is a multifunctional organ responsible for regulation of vascular tone and integrity [14]. The associated endothelial dysfunction is now accepted as a reliable predictor of cardiovascular disease [15]. The assessment of endothelium function is based on endothelium-dependent vasodilation as a response to stimuli, that increases production of endothelium-derived nitrics oxide (NO). Such stimuli include increased shear stress from increased blood flow, (postishemic or driven by receptor-dependent agonists, such as acetylcholine). NO relaxes vascular smooth muscle and is also a potent antioxidant and a regulator of local and systemic redox status [16]. The diagnostic modalities include less common invasive and more frequently used non invasive techiques such as flow-mediated dilatation of the conduit brachial artery using vascular ultrasound [17,18], local vasodilation by venous occlusion plethysmography and microvascular blood flow by laser Doppler flowmetry [19,20], while arterial pulse wave analysis or pulse amplitude tonometry evaluate arterial stiffness which is also highly dependent on fixed structural features of the vascular wall including the degree of fibrosis and calcification [21]. Protocols have been largely standardized and this has resulted in reproducible measurements.
Endothelial dysfunction in T1DM is an early phenomenon that is relatively common even in adolescents with recent onset of diabetes, independent of age, smoking, hypertension, or hyperlipidemia [9]. Higher HbA1c levels in patients with T1DM were associated with more pronounced endothelial dysfunction in the whole population (β = −0.20; P < 0.05) [10], or when focusing only on children and adolescents (β = −0.43; P < 0.01). Moderating impact of BMI on endothelial dysfunction was also found in all individuals with T1DM - the difference in mean BMI between children/adolescents with type 1 diabetes and healthy controls was positively associated with endothelial function but only by analyzing the macrocirculation [10]. Recently it has been postulated, that the main cardiovascular risk factor is not only chronic hyperglycemia or other traditional risk factors, but frequent hypo- and hyperglycemia episodes that accompany the disease daily course - excessive glycemic variability [11,12]. FMD improvement was found (10.9% to 16.6%, p < 0.005) after swiching indvividuals with T1DM to Real-time continuous glucose monitoring (RT-CGM) [13].
Point 4. Line 62-68 in page 2
Why this part contains no reference?
Response 4:
This part has references that are numbered at the end of paragraph [17-21]., it can be indicated next to each individual method. But No. of references will be corrected for purposes of changes in Point 3.
Point 5 All multipple and middle sentence references in all over the manuscript should be reformed.
Response 5
It will be reformed as you suggest in final version as possible, but not in case of listing in the same sentence.
Point 6. Line 72-75 in page 2
This part has no reference. Why?
Response 6.
References will be added as follows:
Most studies that focused on the effect of SGLT2 inhibitors or GLP1 agonists on endothelial function, have been performed in T2DM, and showed discrepant results [22]. Only few of them were performed on T1DM [23]. And for some of those with positive results it was not clear, whether beneficial effects of SGLT-2 inhibitors and GLP-1 RAs are related to direct glucose-lowering effects or other cardiovascular risk factors such as body weight loss and arterial blood pressure modification.
Point 7. About the part "2.1. Study design and subject selection" in line 84
Do the authors perform this part based on previous studies? If yes, which study? And why the have not mentioned the name of this survey?
If no, please explain why you performed this
section in this style?
Response 7.
Study design was not based on previous selected study but was result of many survays that was done about endothelial dysfunction and was mentioned among references in article. Some detailes (for example excluding patients with metformin was decided after results of mentioned reference Lunder et al. [22]. I can mention it in article. This was mentioned also in paragraph about statistic, when the sample size was calculated.
Point 8. Line 199-201 in page 5
This part does not belong to the section "Results". It belongs to the part "material and methods". Please reconsider it.
Response 8.
It will be transformed from Results to Material and methods
Point 9. Why the format of Table 1 is different from Table 2? The format of all tables should be the same and according to the guidline of journal.
Response 9.
Tables will be changed to the same format.
Point 10. About Table 1
Please reform the title of this table. Its vague and does not have enough information to introduce this table to other readers.
Response 10.
The title of table 1 is Baseline caracteristics – why is it vague? Do you suggest to rename it in: Baseline caracteristics and demographic?
Point 11 is missing.
Point 12. line 308 in page 13
Please reform the name "Table 5". It should be "Table 6"
Response 12.
Thank you for noticing the mistake – will be corrected to Table 6.
Point 13. Why the format of tables in this
manuscript is different from each other? Please reform the format of all tables based on the guidline of journal.
Response 13:
The point and the answer is the same as in point 9.
Point 14. About the part "Discussion"
Please rewrite this part according to notes below:
First: categorize all of your results based on their importance (from the most important one to the least important)
Second: after that, turn each one of your results into some subheadings
Third: after that, discuss about them one by one
Forth: make comparisons between your results and the results of other similar and relevant surveys
Response 14:
Newly formed Disscussion
Disscussion
The present study ENDIS revealed that a 12-week course of empagliflozine or semaglutide improves endothelium function in the macro- and microcirculation of patients with T1DM without coexisting cardiovascular or renal diseases when compared to controls.
Improvement in FMD was significant in both intervention groups compared to controls with no changes between those two groups. In arterial stiffness parameters, improvements were seen only in semaglutide group, with significant decline in peripheral resistance. During the evaluation of FBF, which is method for microvascular endothelium function assessment, there were statistically insignificant improvements in both therapeutic groups compared to controls.
In our study, we tried to achieve as comparable groups as possible with respect to glycemic and other metabolic factors, as well as factors that already have a proven impact on endothelial function. Unfortunately, the empagliflozin-treated group and the control group appeared to differ significantly in baseline HbA1C, but after adjustment this difference did not prove to be significant for the final observed events.
Significant reduction in HbA1C seems to have the only strongly predictive value on FMD related outcome, but this is more related to empagliflozin, while the semaglutide effect is independ of this. The outcomes were also not affected by the otherwise non-significant reduction in glucovariability in both treatment groups, nor by the incidence of hypoglycemia, which was very low in all groups. Despite the significant weight loss and reduced waist circumference, indicating a reduction in visceral fat in both treatment groups, this also did not contribute to a beneficial effects which were more favourable in group E .The same can be said for the statistically non-significant reduction in both systolic and diastolic blood pressure, triglycerides and significantly reduction of LDL cholesterol and hs CRP( both seen only in group S) . We can exclude impact of statistically significantly decrease in uric acid in group E, Finally, regarding our study results we can conclude with high confidence, that the beneficial effects of empagliflozin on macrovascular endothelial function are at least partly related to HbA1C reduction while semaglutide effects are more of a reflection of the pleiotropic effect of the drug itself on NO-related dilation, and not simply an effect on glycemic or other metabolic and cardiovascular indices.
Less significant are the results on the second part of the observed macrovascular function - arterial stiffness. However, we observed a reduction in all indicators of arterial stiffness and even a statistically significant reduction in peripheral resistance in those receiving semaglutide. In our study in group E, we observed virtually no difference after the introduction of the drug. The only strong predictor of arterial stiffness improvement is reduction of blood pressure, positive predictor is also LDL reduction and younger age, while glycemic and CRP changes do not have any predictive values.
Although we did not achieve statistical power in the evaluation of reactive stress-induced microvascular hyperaemia, we can conclude, that there is a beneficial effect, which is particularly pronounced with empagliflozin treatment. Empagliflozin had an effect on the flow increase during the control visit already under basal conditions. The flow as a consequence of reactive postischaemic hyperaemia at the control visit (RH 2) compared to the reactive postischaemic flow at the baseline visit (RH 1) was also higher, compared to the control group. And not only was the maximal response increased, but in this group a prolonged response with prolonged hyperaemia was observed even during the 3 min period after cuff release. A slightly lower increase in flow was observed in the semaglutide group, but there was no significant difference compared to the control group. However, compared to empagliflozin, flow was as much as 2-fold lower throughout the 3-minute postischaemic period.
To date, there are few published studies, that have looked at the effect of newer antidiabetic agents on endothelial function, most of which relate to SGLT 2 inhibitors and only a few to GLP 1 agonists, and even among these, semaglutide was not included as an interventional drug. A meta-analysis of 8 studies [22] of the effect of SGLT2 inhibitors on FMD showed, that the ability of SGLT-2i to improve FMD was significant compared to the control group. Those studies suggested that improved vascular function was likely to be associated with empagliflozin‐mediated glycemic and non‐glycemic actions, such as weight loss and volume contraction. All were conducted on T2DM. As they were not adjusted to glycemic outcomes, there was no proven evidence about their efficacy. Another limitation of some of the studies published so far is that they were not randomised to factors that also have a significant impact on endothelial function (e.g. smoking, concomitant statin treatment, ACE inhibitors). In literature statins and ACE inhibitors have beneficial effects on FMD [26,27]. Among the publications, only one is performed in subjects with T1DM and, in this light, is the most comparable to our study [23]. It showed a statistically significant response in endothelial function assessed by FMD in all intervention groups, with the greatest response seen in the group treated with a combination of empagliflozin and metformin, but it was also the only group with a statistically significant reduction in HbA1C, adjustements to glycemic outcomes were not done as was done in our study. Reduction of uric acid was highlighted in the Empa Reg study as one of the possible factors for a better outcome of MACE [1], but uric acid reduction was not associated with FMD outcomes in our study.
Considering the studies published with the effect of SGLT 2 inhibitors on arterial stiffness, the results are controversial. Pooled meta-analysis of five studies showed no significant differences in the change in PWV between the SGLT-2i group and the control group (SMD: 0.11, 95%-CI: − 0.15 ~ 0.37, P = 0.4) [22]. Since the non-significant results were generally in studies with dapagliflozin, experts have formed the opinion [28], that there is a difference between the dapagliflozin and empagliflozin, at least as regards the effect on arterial stiffness and its relevance in long-term cardiovascular studies. This is why we expected more favourable results in group E, as was also seen in a study with a comparative selection of participants which showed improvement of arterial stifness. When compared to placebo, empaglifozin or empaglifozin/metformin signifcantly improved PWV (up to 5.1- and 5.7-fold, both P < 0.01) [23],,. The uncharacteristic values for arterial stiffness parameters in our study may be due to the use of a lower dose of empagliflozin than used in some comparable studies [23,30]. In another study, which compared empagliflozin to placebo, the significant reduction in hs CRP was the only predictor of a reduction in arterial stiffness, in addition to a reduction in blood pressure [29]. We can compare this with our study, strong predictor for arterial stiffness improvement was also reduction of blood pressure, significant reduction of hs CRP was achieved in semaglutide group, but there were no predictive values of such change. In the absence of comparative studies on the effect of semaglutide on this parameter, parallels can be drawn with published studies in T2DM for other GLP 1 agonist agents, which have shown significant reductions in PWV (2 studies; n = 62 patients, pooled MD = −0 18, 95% CI: -0.30 to -0.07, p = 0 002) which were associated with a significant reduction in PWV (pooled MD = −1 97, 95% CI: -2.65 to -1.30, p < 0 001) [31]. In a study investigating the effects of 6-month treatment with the glucagon like peptide-1 analogue liraglutide on arterial stiffness, left ventricular myocardial deformation and oxidative stress in subjects with newly diagnosed type 2 diabetes [32], the groups were equally randomised with respect to all entry parameters, both anthropometric and metabolic, as well as smoking status, ACE I and statin intake. However, markers with an impact on the inflammatory response - e.g. LDL cholesterol or hs CRP were not determined. After adjusting for HbA1c, weight, BMI and waist circumference, patients that received liraglutide had lower PWV, AI, systolic blood pressure and central systolic blood pressure than those on metformin (p < 0.05). Another citated study [33], that was not controlled, examined effects of short-term exenatide treatment on aortic pulse wave velocity of obese T2DM patients, confirmed decline in aortic PWV from 7.2±2.2 m/sec at baseline to 5.1±0.1 m/sec after treatment (P=0.001), there was a significant association with a reduction in waist and hs CRP, but only in younger group of participants. When comparing results of this and our study, better predictor for arterial stiffness improvement is younger age, so we can speculate, that younger patients has more beneficial effect of GLP 1 agonists on arterial stiffness. However, the time of the study was likely too short to be able to draw any major conclusions regarding this matter.
In research of microvascular endothelial assessment in a similar study conducted on T1DM, reactive hyperaemia index (RHI) significantly improved with empaglifozin (up to 1.3-fold, P<0.01) [23]. We could speak of a comparable effect of empagliflozin in our study, although it did not reach statistical power. The methodes of microvascular endothelial function assessment were not the same. In the publications of other SGLT 2 inhibitors, no effect on reactive hyperaemia performed on the microcirculation has shown any significance. There are no comparable studies performed with GLP 1agonists on microvascular endothelial assessment.There have been some promising studies in the past on the effect of GLP 1 agonists on this type of vascular network, in particular using native GLP 1 [34]. There were concusions, that endothelial function is potentially improved by reducing glucose fluctuations through decreasing postprandial hyperglycemia, by reducing postprandial triglycerides levels, and by activating AMPK [35].
Finally, I would like to reiterate, that each antidiabetic drug class could be also associated with differences in their vascular effects. Irrespective of the different correlations, there are certainly synergistic effects of individual metabolic effects on different levels of the vascular network and their vascular response. On the basis of several lines of evidence, we can conclude, that the effect of lowering blood pressure plays an important role in arterial stiffness and peripheral resistance, and also that the effect on the lipid profile takes precedence over the effect of lowering glycemia, which seems to be more important in ensuring the function of the endothelium of large vessels. Even fewer conclusions can be drawn at the microcirculatory level, where none of the changes in metabolic effects appear to have predictive value for improving reactive hyperaemia function. Or, the differences in metabolic outcomes are so small, therefore larger groups of participants would be needed to provide answers. In addition to glycaemic reduction, the reduction of glucose variability also plays an important role, although we could not confirm its importance in our study, but we have to be aware that the population of patients included was extremely well-controlled and, with the use of the CGM system, their glucose variability is also low and not entirely comparable to most of the less well-managed diabetic patients we encounter in our daily practice . Experimental studies have indicated the possibility that dysregulated autoimmune response in T1DM may contribute to endothelial dysfunction by increasing oxidative stress through activation of NADPH oxidase [36]. It still remains unclear whether glucose fluctuations are involved in endothelial dysfunction in patients with type 1 diabetes. All these facts should lead us to strive for better regulation and minimised glucose excursion also with the use of advanced technologies for the treatment of type 1 diabetes.
New antidiabetic drugs empagliflozin and semaglutide are associated with metabolic, cardiovascular and many other pleiotropic effects, therefore are also called diabetes-modulating drugs, because they alter the course and prognosis of the disease. The complexity in understanding endothelial function and influence of new drugs on it was not completely known in T1DM, our study gave us some new answers and prospectives.
This part will be added to Study design: The lower dose was chosen in the study design in view of the equal efficacy and benefit on cardiovascular outcomes of both 10 and 25 mg doses in large CVOT studies performed in T2DM [1] and for safety reasons, to reduce the possibility of DKA. We also took into account the EMA's decision in the case of dapagliflozin, which was the only SGLT 2 inhibitor with short-term approved indication for the treatment of T1DM, the approved dose was lower.
- This is important that the authors compare their findings with the results of prior similar surveys in the part "Discussion".
Response 15:
It was already done in article step by step.
- Please check and adjust the "Reference list" based on the regulations of reference list of journal. (Titles, doi, the name of journal and ... )
Response 16.
It was done according to journal regulations;
Journal Articles:
1. Author 1, A.B.; Author 2, C.D. Title of the article. Abbreviated Journal Name Year, Volume, page range.
It will be checked again and corrected in case of mistakes.
All references will be finally corrected after confirming article changes.
Reviewer 2 Report
This is an original study on the effects of a SGLT2 inhibitor (Empagliflozin) and of a GLP1R agonist (Semaglutide) on the microcirculation of patients with T1DM. It is original, since few studies were conducted on this topic in patients with T1DM (not T2DM). The methods used were broadly adequate. The results are that the SGLT2 inhibitor and the GLP1R agonist improved Flow Mediated Dilation, especially Empagliflozin. The data are of interest.
I have the following comments :
-it is well known that microcirculation is primarly dependent on prevailing blood glucose in T1DM patients (see eg the historical article N Engl J Med 1992 Sep 10;327(11):760-4). Here the investigators did not control glucose with eu- or hyperglycemic clamp. Could they provide data on the actual BG values during the tests ?
-some of the participants were with elevated albumin excretion. This condition is associated with alterations in microcirculation. They should be excluded from the study, or analyzed separately.
-same remark regarding patients with ACE inhibitors or ARBs.
-The SGLT2 inhibitor may have improved FMD through volume reduction. Was there any correlation between changes in FMD and those in haematocrit or haemoglobin in this study ?
-The text especially the discussion is too lenghthy and shoud be divided by two minimally.
-Check Diabetes Duration in Table 1.
Author Response
Response to Reviewer 2 Comments
This is an original study on the effects of a SGLT2 inhibitor (Empagliflozin) and of a GLP1R agonist (Semaglutide) on the microcirculation of patients with T1DM. It is original, since few studies were conducted on this topic in patients with T1DM (not T2DM). The methods used were broadly adequate. The results are that the SGLT2 inhibitor and the GLP1R agonist improved Flow Mediated Dilation, especially Empagliflozin. The data are of interest.
I have the following comments :
Point 1. it is well known that microcirculation is primarly dependent on prevailing blood glucose in T1DM patients (see eg the historical article N Engl J Med 1992 Sep 10;327(11):760-4). Here the investigators did not control glucose with eu- or hyperglycemic clamp. Could they provide data on the actual BG values during the tests ?
Response 1: All measurements and blood takes was done at the same time on the same day at fasting conditions and the blood glucose level was also taken and correlation between results and glucose levels and glucose change between visits was also considered and assessed
Point 2.
some of the participants were with elevated albumin excretion. This condition is associated with alterations in microcirculation. They should be excluded from the study, or analyzed separately.
Response 2: We calculate the correlations and predictive values of outcomes and presence of any of microvascular disease.
Point 3 3.same remark regarding patients with ACE inhibitors or ARBs.
Response 3 We calculate the correlations and predictive values of outcomes and presence of concomitant therapy with ACEi /ARB or statins, which has impact on outcomes.
Point 4.The SGLT2 inhibitor may have improved FMD through volume reduction. Was there any correlation between changes in FMD and those in haematocrit or haemoglobin in this study ?
Response 4. Yes, we measured also hemoglobin . hematocrit , Ht increased for 4,4 in E group and was significantly higher compared to control or to semaglutide group. But in correlation (Spearman coefficient) there were no correlations between Ht change and percent of dilation of FMD (2/1) ; r= 0.172 - weak positive correlation. I added this to text and tables.
|
Correlations |
||||
|
|
∆_Ht |
p_d_2in1 |
||
|
Spearman's rho |
∆_Ht |
Correlation Coefficient |
1,000 |
,172 |
|
Sig. (2-tailed) |
. |
,107 |
||
|
N |
89 |
89 |
||
|
p_d_2in1 |
Correlation Coefficient |
,172 |
1,000 |
|
|
Sig. (2-tailed) |
,107 |
. |
||
|
N |
89 |
89 |
||
Point 5.The text especially the discussion is too lenghthy and shoud be divided by two minimally.
Response 5: I tried to shortened it but there are three main outcomes and three comparable groups and there are many results to discuss so I was not very succesfull in length changes.
Point 6 Check Diabetes Duration in Table 1.
Response 6: Thank you.
Reviewer 3 Report
Maja P.Navodnik and colleagues present a quality and well-written experimental manuscript describing the effect of additional treatment with empagliflozin or semaglutide on endothelial function and arterial stiffness in subjects with type 1 diabetes mellitus (ENDIS Study).
Authors aimed to observe and compare metabolic and endothelial function related effects of newer therapies with good cardiovascular outcomes (empagliflozin and semaglutide) as adjuvant therapies to insulin in well-controlled individuals with T1DM. Most research takes place in T2DM , prescribing of newer therapies is off label with T1DM.
Authors investigated the effect of additional treatment with newer antidiabetic drugs on endothelium function and arterial stiffness in subjects with type 1 diabetes mellitus without cardiovascular diseases. 89 participants, all users of continuous monitoring glucose system, were randomized into three comparable groups, receiving empagliflozin, semaglutide and control group. At baseline and 12 weeks posttreatment authors measured: brachial artery flow-mediated dilation and forearm blood flow as reactive hyperemia assessed with strain gauge plethysmography as parameters of endothelial function, pulse wave velocity, and peripheral resistance as parameters of arterial stiffness.
Authors found that improvement in FMD was significant in both intervention groups compared to controls, with no changes between those two groups. During the evaluation of FBF, there were statistically insignificant improvements in both therapeutic groups compared to controls. In arterial stiffness parameters, improvements were seen only in semaglutide group, with decline in peripheral resistance by 5.1%.
Finally, authors conclude that for arterial stiffness semaglutide seems better, but both drugs positively impact endothelial function, and thus could have a protective role also in T1DM.
Overall, the manuscript is highly valuable for the scientific community and should be accepted for publication.
===========
Other comments:
1) Please check for typos throughout the manuscript.
2) With regards to autoimmune responses - authors are kindly encouraged to cite the following article that describes novel therapeutic targets for autoimmune diseases such as type 1 diabetes mellitus.
DOI: 10.1007/s12668-016-0233-x
Author Response
- Please check for typos throughout the manuscript.
Respond1: I did, thank you.
- With regards to autoimmune responses - authors are kindly encouraged to cite the following article that describes novel therapeutic targets for autoimmune diseases such as type 1 diabetes mellitus.
DOI: 10.1007/s12668-016-0233-x
Respond2:
Do you mean this article?
Bulatov, E., Khaiboullina, S., dos Reis, H.J. et al. Ubiquitin-Proteasome System: Promising Therapeutic Targets in Autoimmune and Neurodegenerative Diseases. BioNanoSci. 6, 341–344 (2016). https://doi.org/10.1007/s12668-016-0233-x
I hope I will read it ( it is not free accessible) – but I don't see correlations between this and our article?
Round 2
Reviewer 1 Report
No more suggestion. Thanks
Reviewer 2 Report
The authors met my concerns.